# Weakly Supervised Scene Graph Grounding

## Abstract

Recent researches have achieved substantial advances in learning structured representations from images. However, current methods rely heavily on the annotated mapping between the nodes of scene graphs and object bounding boxes inside images. Here, we explore the problem of learning the mapping between scene graph nodes and visual objects under weak supervision. Our proposed method learns a metric among visual objects and scene graph nodes by incorporating information from both object features and relational features. Extensive experiments on Visual Genome (VG) and Visual Relation Detection (VRD) datasets verify that our model post an improvement on scene graph grounding task over current state-of-the-art approaches. Further experiments on scene graph parsing task verify the grounding found by our model can reinforce the performance of the existing method.

## 1 Introduction

Motivated by various needs, researchers have designed multiple representations to describe visual contents. More specifically, object bounding boxes localize the objects inside an image while scene graphs represent object-wise interactions. Ideally, each bounding box should corresponds to a node in the scene graph. However, in many cases, such node-object level correspondences are not established, particularly when the information of scene graphs come from non-visual inputs, such as image captions (Wang et al., 2018b), knowledge graph (Zareian et al., 2020b) and commonsense base (Shi et al., 2019).

The lack of node-object level mapping in data results in constraints on various multi-modal learning tasks, e.g., scene graph parsing (Xu et al., 2017; Zhang et al., 2017b), VQA (Ghosh et al., 2019) and image captioning (Yang et al., 2019). If the mapping can be learned without extra annotations, a comprehensive view of an image will be created and benefit a number of downstream tasks. Therefore, in this paper, we focus on grounding scene graph nodes to visual objects under weak supervision, where the node-object correspondences are not annotated even during training phase.

Although the scene graph grounding problem can benefit plentiful downstream tasks, it has been barely studied. Unlike other weakly supervised learning tasks, which focus on single label space (Dietterich et al., 1997; Wang et al., 2018a), the scene graph grounding problem is involved with two label spaces: object categories and relation types, which are disjoint but dependent. More specifically, visual relations are highly correlated with visual objects. As a result, a desirable model should correctly handle the interaction among object categories and visual relations instead of simply learning them independently. Therefore, most of the well-studied weakly supervised learning methods are not suitable for the learning on scene graphs.

Among the few relevant efforts recently spent on this task, Zareian et al. achieve impressive results. They notice the grounding problem when they are trying to handle weakly supervised scene graph parsing. They treat it as a side challenge in weakly supervised learning and propose to tackle it by jointly learning the node-object mapping and a visual relation parser under weak supervision. In their method, a parser capturing the interaction of relation feature and object feature represents the image as a bounding box graph. Then they align such bounding box graph with the scene graph to construct the correspondence between visual regions and scene graph nodes. The mapping found by alignment algorithm is further utilized in optimizing the scene graph parser.

However, the graph aligning process results in one core limitation of their method. Given the fact that the graph-matching problem is NP-hard, they must take the trade-off between efficiency and accuracy into consideration. Furthermore, to enable weakly supervised learning, in training stage

the authors directly generate supervision signal for the parser from the mapping output by the model itself. And as a consequence, an inaccurate graph alignment in initial training stage could mislead the parser, leading to a performance drop.

In this paper, we propose to formulate the mapping problem as a minimum match problem on bipartite graph instead of graph alignment. More specifically, to measure the similarity of a node-object pair, we propose to learn a cost function incorporating information from both object classes and mutual relations. The lack of node-object level mapping brings challenges to the learning process. Therefore, we design an image-graph level distance inspired by Wasserstein Distance(Rüschendorf, 1985), and propose to train our model by minimizing this distance on corresponding image-graph pairs.

Our contributions are:

- This is the first paper that considers weakly supervised scene graph grounding as an independent task and explores it. We argue that weakly supervised scene graph grounding is vital to multi-modal learning and deserves broad attention.

- We propose a novel framework that can bridge visual objects and scene graph nodes by capturing both object classes and relational information and searching the best match among the objects and nodes. By designing an image-graph level distance, we tackle the challenge brought by the lack of supervision.

- Empirical results indicate that our grounding method outperforms existing mapping methods. Moreover, we also verify the value of our model in enhancing the performance of existing SOTA models on weakly supervised scene graph parsing, an important and beneficial downstream task in multi-modal learning.

## 2 RELATED WORK

**Visual Relation Detection (VRD):** The goal of VRD is to further understand an image by detecting the relations between detected object pairs. Powered by datasets consisting of visual relation annotations, e.g., Visual Relation Detection (Lu et al., 2016) and Visual Genome (VG) (Krishna et al., 2017), substantial advancements are made by recent researches. Several convolutional neural network (CNN) architectures (Zhang et al., 2017b; Yin et al., 2018; Li et al., 2017; Inayoshi et al., 2020) are tailored for recognizing visual relations, while some other researches (Dai et al., 2017; Zhang et al., 2017b; Mi & Chen, 2020) focus on post-CNN feature inference. Apart from the vision domain, some researchers (Lu et al., 2016; Yu et al., 2017; Zhang et al., 2017a) pay their attention to the language domain, employing linguistic priors for better predicate predictions. Recent researches (Baldassarre et al., 2020; Peyre et al., 2017) explore the problem with weak supervision.

**Scene Graph Parsing:** Unlike the VRD task, scene graph parsing, first introduced by (Xu et al., 2017), aims at parsing a structured representation of the given image. A few works (Tang et al., 2019; Wang et al., 2019; Lin et al., 2020; Zellers et al., 2018) parse scene graphs by reasoning over visual context. Furthermore, Yang et al. proposes an R-CNN (Girshick, 2015) like model with additional graph convolutional networks (GCN) (Kipf & Welling, 2017). Additionally, real-world commonsense, as well as external knowledge, are utilized to guide the scene graph parsing process (Zareian et al., 2020b; Gu et al., 2019; Chen et al., 2019; Zareian et al., 2020c). Recently, Zareian et al. proposes a novel graph alignment to parse scene graphs under weak supervision.

**Graph Neural Networks (GNN):** Recent years, a great quantity of research efforts have been devoted into Graph Neural Networks (GNN) (Scarselli et al., 2008; Bruna et al., 2013; Kipf & Welling, 2016). By adopting suitable aggregation strategies, GNN can capture features from various aspects, including edges, nodes and the whole graph (Battaglia et al., 2018). In computer vision fields, graph neural networks can be applied to capture the structural information on both images and scene graphs (Yang et al., 2018; Johnson et al., 2018; Yang et al., 2019). More specifically, by aggregating structural information on the input graph, GNN can represent a scene graph node or a visual object as an embedding vector containing relational information of the edges connected to it.

**Weakly Supervised Learning (WSL):** Weakly supervised learning has been utilized in many computer vision tasks like object detection, semantic segmentation and visual relation detection (Zhang et al., 2017b; Huang et al., 2018; Zareian et al., 2020a). Many of them belong to Multiple Instance

Learning (MIL)(Dietterich et al., 1997; Maron & Lozano-Pérez, 1998). When applying neural networks in multiple instance learning, instance-level pooling (Liu et al., 2012) and feature-level pooling(Wang et al., 2018a) are two common architectures in practice, making MIL compatible with gradient-based optimization strategies.

## 3 TASK DEFINITION

A scene graph is a structured representation of an image, capturing both visual objects and their relations. We denote an image as $I$, regarded as a set of visual objects $\{v_1, v_2, ..., v_m\}$. We assume that the features and locations of visual objects can be extracted by a pre-trained object detector. [1] And the scene graph describing $I$ can be formulated as $G = \langle U, R \rangle$, where $U = \{u_1, u_2, ..., u_n\}$ is the node set ($n \leq m$) and $R = \{r_1, r_2, ..., r_k\}$ is the edge set. Each node $u$ has a label $y_u$ which describes the object category it refers to. And each edge $r$ is represented as a triplet $\langle u_i, u_j, y_r \rangle$, where $u_i$ is the source node, $u_j$ is the target node and $y_r$ is the relation type.

**Scene Graph Grounding:** Though the graph-image pair $\langle G, I \rangle$ can be easily retrieved, the correspondences between graph nodes $U$ and visual objects $V$ remain unclear. The goal of scene graph grounding is to map each node $u$ to a visual object $v$ inside the image.

**Weakly Supervised Scene Graph Grounding:** As manually labeling the ground-truth node-object references is expensive and time consuming, we aims at learning a grounding model without relying on such annotations. In *Weakly Supervised Scene Graph Grounding*, we are given a training set $D$, where the each sample in $D$ is a tuple of an image $I$ and its scene graph representation $G$. However, the ground truth node-object mapping is not provided for any tuple. We aims at learning a grounding model from $D$.

## 4 PROPOSED METHOD

To build up the mapping between visual objects and scene graph nodes, we propose to learn a object-node wise metric that incorporates information from both object categories and relations. Given an ungrounded image-graph pair, we measure the similarity of each object-node pair using learned metric and build a bipartite graph among the object set and the node set. In this way, we formulate the object-node mapping problem as searching the *minimum weight match* in a bipartite graph. Unlike the NP-hard graph matching problem in previous work (Zareian et al., 2020a), our formulation leads to a polynomial time solution. The overview of our framework is shown in Fig. 1. Formally, we define the metric on a given object-node pair $(u, v)$ as following cost function:

$$c(u, v) = \lambda c_{obj}(u, v) + (1 - \lambda)c_{rel}(u, v) \tag{1}$$

where $c_{obj}(u, v)$ is the object-class based metric and $c_{rel}(u, v)$ refers to the relation based metric.

### 4.1 OBJECT-CLASS BASED METRIC

Given a scene graph node $u$ and a visual object $v$, we define following metric to incorporate object category information:

$$c_{obj}(u, v) = P(y_u \neq y_v) = 1 - P(y_u = y_v) \tag{2}$$

where $y_u$ is the object category of node $u$ and $y_v$ is the category of $v$. Since the ground-truth object-node mapping is not accessible during training phase, we can not acquire the ground truth categories of visual objects. Therefore, we train a neural network to estimate $P(y_u = y_v)$ through pool based multiple instance learning. The estimate is denoted as $\hat{P}(y_u = y_v; \theta_{obj})$ where $\theta_{obj}$ is the parameter of the network. We optimize the neural network with the following loss function:

$$\mathcal{L}_{MIL}(G, I; \theta_{obj}) = - \sum_{y \in Y(U)} \log \sigma(\{\hat{P}(y_v = y; \theta_{obj})|v \in I\}) \tag{3}$$

where $I$ is the image, $U$ is the node set of corresponding scene graph $G$, $Y(U)$ is the object category set of $U$ and $\sigma$ is a pooling operator. For a class $y$ existing in the image, the pooling operator aggregates the probabilities $\hat{P}(y_v = y; \theta_{obj})$ over all objects $v$ in the image $I$. It can be any

---

[1]However, due to the label space difference between pre-training dataset and scene graph dataset, we can not give the category label of the visual objects

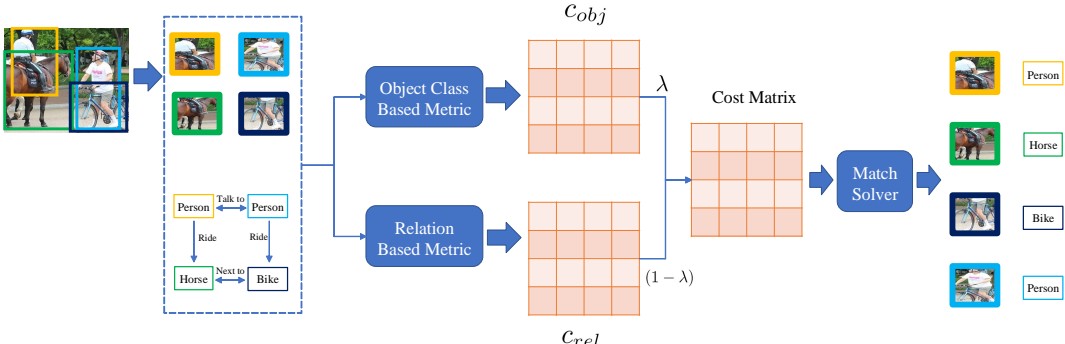

Figure 1: Overview of Our Framework: Bounding boxes and RoI features of visual objects are extracted by a pretrained detector; The object class based metric and relation based metric compute the cost matrix between visual objects and scene graph nodes; A matching algorithm is utilized to predict the grounding results.

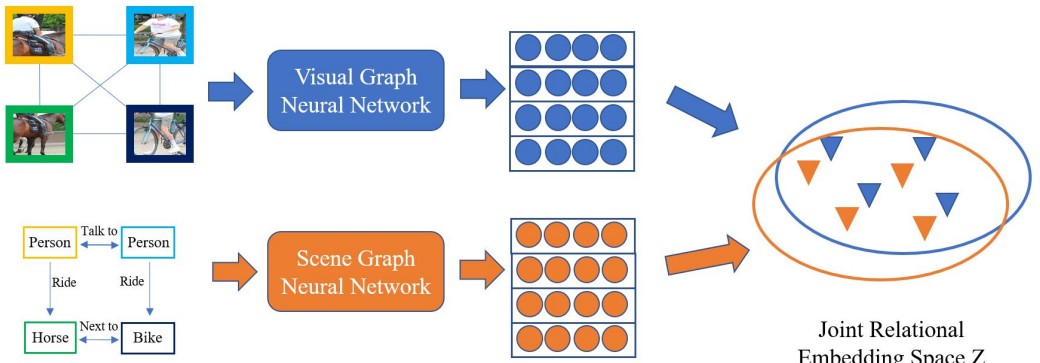

Figure 2: Overview of Our Relation based Metric Module: Visual objects and scene graphs are fed into two independent GNN model, which map them to the joint relational space $Z$.

permutation-invariant operator, such as $\max$ or mean. With the learned $\hat{P}(y_u = y_v)$, we define $c_{obj}$ as:

$$c_{obj}(u, v; \theta_{obj}) = 1 - \hat{P}(y_u = y_v; \theta_{obj}) \tag{4}$$

## 4.2 RELATION BASED METRIC

Apart from object category, the object-wise interaction and relation-object interaction are also important for learning the mapping, as the scene in an image may be complicated (He et al., 2020). For example, in one image there might be a "person" "riding" a "horse" and a "person" "riding" a "bike". It is crucial to correctly distinguish them for some downstream tasks like weakly supervised scene graph parsing.

To learn a metric capturing above interaction information from both visual input and scene graph, we propose a two-stream architecture, as shown in Fig. 2, to model the structural similarity. The two streams are two graph neural networks that project visual objects and scene graph nodes into a shared embedding space $Z$. In visual stream, to capture object-wise interaction, we first construct a fully connected graph where each node is a visual object. Then the graph is forwarded into a graph neural network, where the $i$-th layer updates the representation of object $v$ as follows:

$$h_u^{i+1} = \mathrm{MLP}_2(\mathrm{AGGREGATE}(\{\mathrm{MLP}_1(h_u^i \oplus h_{u'}^i) | u' \in \mathcal{N}(u)\})) \tag{5}$$

where $\oplus$ means concatenate, AGGREGATE is an aggregation operator, and $\mathcal{N}(u)$ is the neighbor set of $u$. We initialize the representation of each visual object as the features extracted by a pre-trained visual-encoder. Meanwhile, to capture the relation features in scene graph, we forward it into another graph neural network proposed in Johnson et al. (2018), which can model the relation-object interactions. The initial representation of a node or an edge is the word embedding of its class name. Its updating rule in each layer is shown in Alg. 1.

---

**Algorithm 1** Updating algorithm of Scene Graph Neural Network.

---

**Require:** A scene graph $G = \langle U, R \rangle$, node representation $h_u$ of each node $u \in U$ and edge representation $h_r$ of each edge $r \in R$
**Ensure:** Updated node representation $h'_u$ and edge representation $h'_r$.
1: **for** each edge $r = \langle s, o, y_r \rangle$ **do**
2:      $h_t = g_r(h_s \oplus h_r \oplus h_o)\{g_r$ is a function aggregating information from a triplet$\}$
3:      $h'_r = f_r(h_t)\{f_s$ is a linear layer with activation$\}$
4: **end for**
5: **for** each node $u$ **do**
6:      $h^s_u = \text{AGGREGATE}(\{f_s(h_t)|r \in \mathcal{S}(u)\})$ $\{f_s$ is a linear layer with activation and $\mathcal{S}(u)$ is the set of edges whose source node is $u\}$
7:      $h^o_u = \text{AGGREGATE}(\{f_o(h_t)|r \in \mathcal{O}(u)\})$ $\{f_o$ is a linear layer with activation and $\mathcal{O}(u)$ is the set of edges whose target node is $u\}$
8:      $h'_u = \text{MLP}(\text{AGGREGATE}(h^s_u, h^o_u))$
9: **end for**

---

Given a scene graph node $u \in G$ and a visual object $v \in I$, let us denoted their representation in space $Z$ as $h(u)$ and $h(v)$ respectively. Then we can calculate the structural similarity between $u$ and $v$ as:

$$c_{rel}(u, v; \theta_{rel}) = d(h(u), h(v)) \tag{6}$$

where $d$ is a vector metric on space $Z$ and $\theta_{rel}$ is the parameters in relational metric module. Since graph neural network can represent structural information on graphs as vectors, by comparing $h(u)$ and $h(v)$, we can measure the relational similarity of an object-node pair. However, training the graph neural networks is very challenging. Because when ground truth object-node mapping is not accessible, we can not simply minimize the distance of mapped object-node pairs. Since graph structure can not be simply classified to a set of discrete categories, we can not apply similar strategy as in learning object category metric.

Inspired by Wasserstein Distance (Rüschendorf, 1985), defined as the cost expect of the optimal transport on two distributions, we propose a distance that applicable to two discrete sets. Specifically, we apply the cost sum of the optimal linear sum assignment on the node sets and object sets as image-graph level distance, denoted as $D(G, I)$:

$$D(G, I) = \sum_{(u,v) \in M^*} c(u, v) = \sum_{(u,v) \in M^*} \lambda c_{obj}(u, v) + (1 - \lambda)c_{rel}(u, v) \tag{7}$$

where the definition of $M^*$ is

$$M^* = \underset{M \in \pi(G,I)}{\arg\min} \sum_{(u,v) \in M} c(u, v) \tag{8}$$

where $\pi(G, I)$ is the set of all possible match between the scene graph node set and visual object set. We then propose to train the model by minimizing $D(G, I)$ for all mapped image-graph pairs.

Precisely optimizing above distance via gradient descent is intractable, because the process of searching the minimum match $M^*$ is non-differential. An alternative solution is that in each iteration, we first calculate the minimum match $M^*$ based on current GNN parameters, then run a gradient descent step that minimizes the distance of matched node-object pairs.[2] However, in the initial training stage, the relational embeddings randomly spread in the space. Thus, the match found

---

[2]The justification of this alternative solution can be found in Appendix

under current parameter may contain noise and mislead the training. To address this challenge, we approximate $M^*$ as $M'$, which is defined as:

$$M' = \arg\min_{M \in \pi(G,I)} \sum_{(u,v) \in M} c_{obj}(u,v). \tag{9}$$

Since searching the minimum match on bipartite graph can be solved by Hungarian Algorithm (Kuhn, 1955) in polynomial time, such approximation can be computed as long as the multiple instance classifier is trained in advance. Our loss function on graph neural network branch can be written as:

$$\mathcal{L}_{gnn}(G,I) = \sum_{(u,v) \in M'} \lambda c_{obj}(u,v) + (1-\lambda)c_{rel}(u,v) = C + (1-\lambda) \sum_{(u,v) \in M'} \lambda c_{rel}(u,v) \tag{10}$$

where $C$ is a constant. Note that if we apply simple Euclidean distance or inner-product distance as $d$, above loss function will have a trivial solution where $c_{rel}(u,v) = 0$ for all $(u,v)$. To avoid it, we use their normalized version. Given a distance $d$ like Euclidean distance, its normalized version is defined as:

$$d_{norm}(u,v) = -\frac{\exp\left(-d(u,v)\right)}{\sum_{k \in I} \exp\left(-d(u,k)\right)} \tag{11}$$

### 4.3 TRAINING AND INFERENCE

The training algorithm of our model is illustrated in Alg. 2. In general, the training can be divided as two phases. In the first phase, we train the object-class based metric via pool based multiple instance learning. In the second phase, we train the Graph Neural Networks based on the well-trained object-class based metric. In testing stage, since $\theta_{obj}$ and $\theta_{rel}$ have been well-trained, we can calculate $M^*$ with Hungarian Algorithm simply and output $M^*$ as mapping result.

---

**Algorithm 2** Training Algorithm on Weakly Supervised Dataset $D$.

---

**Require:** Dataset $D$
**Ensure:** The well-trained parameters $\theta_{obj}$ of the object category based metric and $\theta_{rel}$ of the relation based metric.
1: Randomly initialize $\theta_{obj}$ and $\theta_{rel}$
2: **while** $\theta_{obj}$ not converged **do**
3:     Sample a batch of samples $(I, G)$ from $D$ and forward the samples into the neural networks in Sec. 4.1.
4:     Calculate the loss function $\mathcal{L}_{MIL}$ in Eq. 3
5:     Update $\theta_{obj}$ according to the loss function $\mathcal{L}_{MIL}$.
6: **end while**
7: Calculate minimum match $M'$ with Hungarian algorithm
8: **while** $\theta_{rel}$ not converged **do**
9:     Sample a batch of samples $(I, G)$ from $D$ and forward the samples into the graph neural networks in Sec. 4.2.
10:     Calculate the loss function $\mathcal{L}_{gnn}$ in Eq. 10
11:     Update $\theta_{obj}$ according to the loss function $\mathcal{L}_{gnn}$.
12: **end while**

---

## 5 EXPERIMENTS ON WEAKLY SUPERVISED SCENE GRAPH GROUNDING

### 5.1 EXPERIMENT SETTING

**Datasets:** We evaluate our grounding approach on two prevailing datasets: *Visual Genome (VG)*(Krishna et al., 2017) and *Visual Relation Detection (VRD)* (Lu et al., 2016). For VG dataset, we apply the split and preprocess protocol in Xu et al. (2017). For VRD dataset, we apply the original split and protocal in the original paper (Lu et al., 2016).

**Baselines:** We compare our method with following baselines, which support weakly supervised learning on images and their ungrounded scene graphs: (1) *VSPNet* (Zareian et al., 2020a) is a

| Methods | Visual Genome | | VRD | |
|---|---|---|---|---|
| | Ground Truth | RPN Proposal | Ground Truth | RPN Proposal |
| WS-VRD | 36.0 | 10.7 | 27.8 | 9.6 |
| VSPNet-v1 | 38.6 | 14.7 | 52.6 | 9.7 |
| VSPNet-v2 | 41.9 | 15.2 | 57.7 | 13.1 |
| Our Method | **50.8** | **15.8** | **68.9** | **16.5** |

Table 1: Results of Weakly Supervised Scene Graph Grounding Methods with VG and VRD: Our proposed model surpass all baseline models by noticeable margins.

framework jointly learning scene graph parsing and scene graph grounding under weak supervision. Its graph alignment module can alternatively select starting from mapping edges or mapping nodes. We denote the first one as *VSPNet-v1* and the latter one as *VSPNet-v2*; (2)*WS-VRD* (Baldassarre et al., 2020) is an explanation-based weakly supervised scene graph parser. It first aggregate the relation feature of all visual object pairs and predict all possible relation types that may exist in the image. Then it assign each predicted relationship label to the object pair that make the greatest contribution to the prediction.

Among models above, *WS-VRD* does not have an explicit grounding module. Therefore, to evaluate its performance on grounding task, we borrow the idea of VSPNet and first use it to transform input images to scene graphs where each node is a bounding box. Then we use a graph alignment algorithm to map each scene graph node to a bounding box based on the output graph.

**Visual Object Extraction:** To extract the visual object features from raw image, following VSPNet (Zareian et al., 2020a) we apply an off-the-shelf Faster-RCNN (Ren et al., 2015) object detector pre-trained on OpenImage (Kuznetsova et al., 2020) dataset. The location of each object is represented as a bounding box. Since scene graph grounding is highly affected by the quality of object proposals, we report grounding accuracy under two settings: one is that the object detector is given ground truth bounding box as proposal, denoted as *Ground Truth*, and the other is that the object detector use the proposals from its own RPN (Region Proposal Network) module, denoted as *RPN Proposal*. To be fair, we apply this extraction protocol on all methods.

**Evaluation Metrics:** We apply *Accuracy* to evaluate the models on grounding task following Sadhu et al. (2019). In *Ground Truth* setting, a node is correctly mapped if it is exactly mapped to the corresponding ground truth bounding box. In *RPN Proposal* setting, a node is correctly mapped if the IoU score between the mapped box and ground truth box is larger or equal to 0.5. The final accuracy is averaged over all scene graph nodes.

## 5.2 RESULTS ON SCENE GRAPH GROUNDING

Quantitative results shown in Tab. 1. illustrates our advantages on both datasets, regardless of which type of proposals are given. On Visual Genome, our model improves the recall by nearly 9% with ground truth bounding boxes, while gets a slightly higher recall with RPN proposals. Besides, our model achieves 68.9% recall, more than 10% higher than any other baselines, on VRD with ground truth bounding boxes. Additionally, with proposals generated by RPN, our model shows a 3% boost on the recall. It is noticeable that the improvement made by our proposed method is much larger when ground truth bounding boxes are given. Such difference indicates that our model can better learn the mapping between objects and scene graph nodes with desirable object proposals.

## 5.3 ABLATION STUDY

To quantify the contribution of different components of our model, we conduct ablation study by comparing our model with two variants: (1) Without Minimum Match: after calculating the cost function matrix, instead of conducting minimum match via Hungarian Algorithm, we simply map each node to the box with minimum cost function value with it; (2) Without Relational Metric: we set $\lambda$ in Eq. 1 to 1 so that the model only makes use of the object category metric.

The results are shown in Tab. 2. Compared with the model without minimum matching, our model has a noticeable performance boost, indicating that matching node-box through Hungarian algo-

| Methods | Visual Genome | | VRD | |
|---|---|---|---|---|
| | Ground Truth | RPN Proposal | Ground Truth | RPN Proposal |
| Ours w/o minimum matching | 40.6 | 12.1 | 57.2 | 10.5 |
| Ours w/o relational metric | 49.1 | 15.4 | 66.7 | **16.6** |
| Ours (full) | **50.8** | **15.8** | **68.9** | 16.5 |

Table 2: Ablation Study on Our Grounding Model: The results show that our model benefits from the matching process and the relational metric

| Node Type | Node Numbers | Acc w/o rel-metric | Acc with rel-metric |
|---|---|---|---|
| Ambi-Nodes with relations | 106k | 28.3 | 33.3 |
| Ambi-Nodes without relations | 91k | 26.0 | 26.9 |
| Unique Nodes | 155k | 77.2 | 77.3 |

Table 3: Detailed Ablation Study on Our Grounding Model on VG dataset: The results show that our model benefits from the matching process and the relational metric

rithm is much better than greedy methods. It is also shown that our model is able to learn feasible information from data, compared with the model without relational metrics.

Furthermore, to verify that our relational based metric successfully learn to use relational information to distinguish the nodes with same object class (like the case of "person riding horse" and "person riding bike"), we report the grounding accuracy on different fractions of the nodes on test set of Visual Genome Dataset. We apply the ground truth bounding boxes as candidate boxes. We divide the nodes to three types: ambiguous nodes with relations, ambiguous nodes without relations and unique nodes. A node is an unique node if there is no other nodes belonging to the same object class of it in the same graph. Otherwise, the node is an ambiguous node. Obviously, for unique nodes, the object class information is adequate for grounding. For ambiguous nodes with relations, relational information is important for distinguishing them from the nodes with same object class.

The results shown in Tab. 3 demostrates that, for the ambiguous nodes with relations, the relational metric significantly improve the grounding accuracy. Also, even for the ambiguous nodes that do not have relations, relational metric also helps the grounding. This is because our model applies minimum matching. When matching the ambiguous nodes with relations correctly, it also help the ambiguous nodes without relations exclude wrong options. Meanwhile, our relational metric does not interrupt the grounding of the unique nodes, which can be well handled by the object-based metric.

# 6 APPLICATION: IMPROVING WEAKLY SUPERVISED SCENE GRAPH PARSING

Apart from scene graph grounding shown in Sec. 5, we further apply our grounding results to a downstream task: weakly supervised scene graph parsing. More specifically, given a weakly supervised scene graph parsing dataset, we first apply our grounding model to map each node in the scene graph to a bounding box in the image. In this way we generate grounded scene graph parsing annotations. We use above annotations to train scene graph parsers in fully supervised manner. In the evaluation stage, we compare them with the parsers directly trained on the original dataset under weak supervision.

**Experiment Setting:** We conduct this experiment on Visual Genome with three weakly supervised methods: (1) *VSPNet* (Zareian et al., 2020a), the current state-of-the-art; (2) A multi-layer perceptron (*MLP*) which accepts the concatenation of two object features as input and predicts their visual relations; (3) A graph neural network (*GNN*) over fully connected graphs where each node represents one visual object. All three models can be trained under full supervision or weak supervision. To train the MLP and the GNN under weak supervision, we adopt the multi-instance-learning (MIL) based loss function.

| Methods | Supervision | SGCls | | PredCls | |
|---|---|---|---|---|---|
| | | Recall@50 | Recall@100 | Recall@50 | Recall@100 |
| MLP FS | Full | 22.8 | 25.9 | 40.7 | 49.3 |
| GNN FS | Full | 23.5 | 26.6 | 41.5 | 50.0 |
| VSPNet FS | Full | **31.5** | **34.1** | **67.4** | **73.7** |
| MLP-MIL | Weak | 19.7 | 22.5 | 38.2 | 47.0 |
| GNN-MIL | Weak | 18.2 | 20.7 | 37.4 | 45.8 |
| VSPNet WS | Weak | 30.5 | 32.7 | 57.7 | 62.4 |
| MLP (Ours) | Weak | 22.4 | 25.1 | 41.1 | 49.8 |
| GNN (Ours) | Weak | 21.7 | 24.8 | 38.8 | 47.8 |
| VSPNet (Ours) | Weak | **30.6** | **33.3** | **59.9** | **65.2** |

Table 4: Results of Scene Graph Classification and Predicate Classification on VG: Our grounding results can act as a reinforcement to other scene graph parsing models and further improve their performances

| Methods | SGGen | |
|---|---|---|
| | Recall@50 | Recall@100 |
| MLP-MIL | 1.9 | 2.2 |
| GNN-MIL | 1.5 | 1.8 |
| VSPNet WS | 4.7 | 5.4 |
| MLP (Ours) | 2.4 | 3.0 |
| GNN (Ours) | 2.4 | 3.0 |
| VSPNet (Ours) | **5.1** | **5.8** |

Table 5: Results of Scene Graph Generation on VG: Our grounding results can bring boosts to other scene graph parsing models in the scenario without ground-truth bounding boxes

**Evaluation Metrics:** Following (Herzig et al., 2018) and (Zareian et al., 2020a), we evaluate the models above with three setups: scene graph generation (SGGen), scene graph classification (SGCls) and predicate classification (PredCls). SGGen requires a model to generate triplets from an input image without ground truth bounding box. A generated triplet is considered as correct if (1) for both subject and object, the detected bounding boxes have an IoU of at least 0.5 with ground truth and (2) the categories of subject, object and relation are all correctly predicted. SGCls requires a model to predict object categories along with their mutual relations given ground truth bounding boxes. And PredCls asks for predicting relations for all visual object pairs given ground truth bounding boxes. We report Recall@K score, the ratio of ground truth triplets correctly detected by the model when top K triplet predictions are taken into account, with K=50, 100. For the fully supervised models, we only report their performance of SGCls and PredCls because they require denser proposals (300) in SGGen setup compared to the weakly supervised models (20 proposals required), which is unfair for the comparison.

**Results:** Quantitative results are shown in Tab. 4 and 5. Such results demonstrate that our grounding model can act as an reinforcement for other weakly supervised approaches. In particular, our grounding improves the performance of the MLP model for nearly 3% over all 4 metrics with ground truth boxes. The recalls of the GNN model also raise for more than 1% with the help of our grounding results. Additionally, the correspondences generated from our model brings performance boost to VSPNet, the state-of-the-art, on all metrics of all set up.

# 7 CONCLUSION

In this paper, we propose the task of weakly supervised scene graph grounding, establishing correspondences between visual objects and scene graph nodes, and provide an algorithm solving it. Our method surpasses the current state-of-the-art on the task. Furthermore, the outputs of our model can be applied to downstream scene graph parsing algorithm and effectively improve the performance.

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

APPENDIX

## A   POOLING-BASED MIL

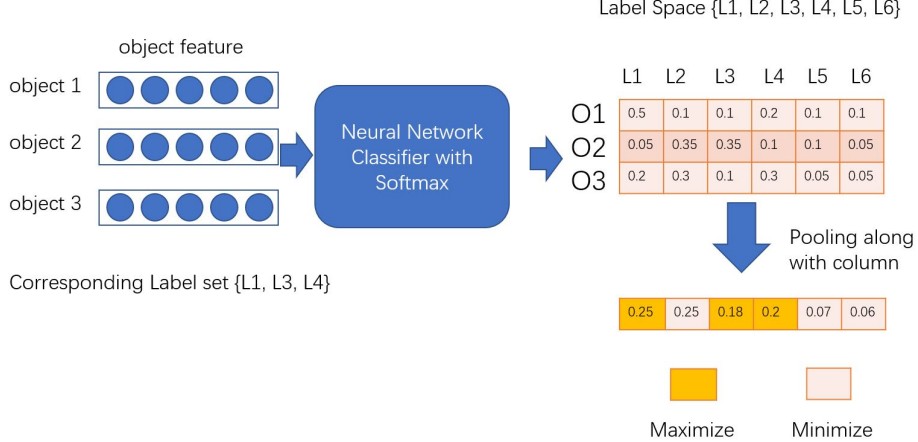

Figure 3: The pipeline of pooling-based MIL.

Here we describe the pooling-based MIL with a pipeline in Figure 3. In each updating iteration, a neural network classifier takes the object features in and output the probabilities of each label in the label space. The probability vectors form a matrix where each raw corresponds to an object. After that, a pooling operator like mean aggregates each column in the matrix to a scalar. Then a gradient descent step is applied to maximize the aggregated probability of labels existing in the image and minimize the others.

## B   JUSTIFICATION OF THE ALTERNATIVE OPTIMIZATION METHOD

Here we give the justification of the alternative optimization method mentioned in Sec 4.2.

Let $c(u, v; \theta_{obj}, \theta_{rel})$ represent the cost function given current model parameters $\theta_{obj}, \theta_{rel}$. Let us define $M^*(\theta_{obj}, \theta_{rel})$ as:

$$M^*(\theta_{obj}, \theta_{rel}) = \underset{M \in \pi(G,I)}{\arg\min} \sum_{(u,v) \in M} c(u, v; theta_{obj}, \theta_{rel}) \tag{12}$$

Suppose we run an optimization step and obtained new parameters $\theta'_{obj}, \theta'_{rel}$. If the new parameters satisfy:

$$\sum_{(u,v) \in M^*(\theta_{obj}, \theta_{rel})} c(u, v; \theta_{obj}, \theta_{rel}) \geq \sum_{(u,v) \in M^*(\theta_{obj}, \theta_{rel})} c(u, v; \theta'_{obj}, \theta'_{rel}) \tag{13}$$

Based on the definition of $M^*(\theta'_{obj}, \theta'_{rel})$ we have:

$$\sum_{(u,v) \in M^*(\theta_{obj}, \theta_{rel})} c(u, v; \theta'_{obj}, \theta'_{rel}) \geq \sum_{(u,v) \in M^*(\theta'_{obj}, \theta'_{rel})} c(u, v; \theta'_{obj}, \theta'_{rel}) \tag{14}$$

And thus we have:

$$\sum_{(u,v) \in M^*(\theta_{obj}, \theta_{rel})} c(u, v; \theta_{obj}, \theta_{rel}) \geq \sum_{(u,v) \in M^*(\theta'_{obj}, \theta'_{rel})} c(u, v; \theta'_{obj}, \theta'_{rel}) \tag{15}$$

## C   DATASETS

As mentioned in Sec. 5, we conduct our experiment with two datasets *Visual Genome (VG)* (Krishna et al., 2017) and *Visual Relation Detection(VRD)* (Lu et al., 2016).

- *VG* comes with 108k images, along with object, relation and scene graph annotations. Following Zareian et al. (2020a), we keep 150 most frequent entities categories and 50 most frequent predicate classes. We also follow the split adopted by Zareian et al. (2020a), which contains 75k images for training and 32k for testing.
- *VRD* contains 5k images with 100 object categories and 70 predicate types. More specifically, the training set consists of 4k images while the test set has 1k.

## D  IMPLEMENTATION DETAILS

### D.1  OVERALL SETTING OF GROUNDING MODEL

We set $\lambda = 0.5$ for VG dataset and $\lambda = 0.9$ for VRD datasset. As for the word embedding, in our implementation, for VG dataset, we apply the same pre-trained GloVe word embedding (Pennington et al., 2014) as VSPNet for both object category names and relation type names. For VRD dataset, we apply the word embedding in spacy library. The word embedding dimensions of all datasets and methods are set as 300, following VSPNet. We apply Adam (Kingma & Ba, 2014) with learning rate of $1e - 4$ to optimize our model

### D.2  OBJECT CLASS BASED METRIC

Here, let use denote the object category label set of the dataset as $L = \{l_1, l_2, ...\}$. We apply two-layer perceptron to estimate $P(y_v|v)$ given the feature extracted by the Faster-RCNN. In the case where ground-truth bounding boxes are given, we directly use the perceptron to predict the category of the object. More specifically, we set the output dimension of the percepetron as $|L|$, which is the object category number of the dataset (150 in VG and 100 in VRD). Then we apply Softmax function to normalize the output vector to be a distribution. In the case where we are given RPN proposals, we first use the perceptron to project an object feature to a 300-dimension vector $x_v$. Then we calculate the inner dot product between the 300-dimension vector and word embedding of each object category. Then the object category distribution $P(y_v|x_v)$ of an object $v$ is predicted as:

$$P(y_v = l_i|x_v) = \frac{\exp x_v \cdot w(l_i)}{\sum_{j<|L|} \exp x_v \cdot w(l_j)} \tag{16}$$

where $w(l_i)$ is the word embedding of category $l_i$. For the pooling operator $\sigma$ in $\mathcal{L}_{MIL}$, we apply mean for the RPN proposal setting of VG and $\max$ for others.

#### D.2.1  RELATIONSHIP BASED METRIC

**Visual Graph Neural Network:** We apply mean as aggregating operation. We set the $\text{MLP}_1$ and $\text{MLP}_2$ as two-layer perceptrons, with hidden dimension of $512$ and output dimension of $512$.

**Scene Graph Neural Network:** We apply sum as aggregating operator in line 6 and 7 of Alg. 1. As for the aggregating operator in line 8, we apply following definition:

$$\text{AGGREGATE}(h_u^s, h_u^o) = \frac{h_u^s + h_u^o}{|\mathcal{S}(u)| + |\mathcal{O}(u)|} \tag{17}$$

which is the same as Johnson et al.. For the MLP in line 8, we apply a two-layer perceptron, with hidden dimension of $512$ and output dimension of $512$.

#### D.2.2  SETTING OF SCENE GRAPH PARSING MODEL

Here we introduce the implementation of MLP and GNN parsing models in detail. As for the VSPNet, we directly apply all of the original setting in the original implementation of the authors.

**Loss function:** The loss function of MLP FS and GNN FS is cross entropy applied on object category and relation type respectively. As for MLP-MIL and GNN-MIL, the loss function of object category is the same as the loss function of object class based metric. And the loss function of relation type on one sample $\langle I, G = \langle U, R \rangle \rangle$ is defined as:

$$\mathcal{L} = \sum_{\langle u,u',y_r \rangle \in R} \log \sigma(\{P(y_v = y_u)P(y_{v'} = y_{u'})P(y_{(v,v')} = y_r)|v, v' \in I\}) \tag{18}$$

where $\sigma$ is a max-pooling operator among all possible object pairs. $P(y_{(v,v')} = y_r)$ is the possibility that the relation from $v$ to $v'$ belongs to type $y_r$ predicted by the model. When applying our grounding results on MLP, GNN and VSPNet, we just treat our grounding result as ground-truth and run the model in fully supervisd setting.

**Optimization:** We apply Adam (Kingma & Ba, 2014) with learning rate of $1e - 4$ to optimize our model. We use a validation set of 1k images for early stopping. To be fair, for all three settings (fully supervision, weakly supervision and the version with our grounding), we provide ground-truth scene graph on the validation sets for GNN and MLP. As for VSPNet, we directly copy their hyperparameter setup. In VSPNet + our Grounding setting, we borrow the hyperparameters of their fully supervised version.

**Neural Network Architecture:** For MLP, we apply two-layer perceptrons whose hidden dimension number equal to their input dimension number. For GNN, we directly use the architecture and hyperparameter of the Visual Graph Neural Network in grounding model. We concatenate the output of the VGNN and the initial feature from Faster-RCNN to get the final feature of each visual object. Then we feed them into a MLP to predict their categories and feed the concatenation of all object pairs into another MLP to predict their relation categories.

## E   VISUALIZATION OF OUR GROUNDING MODEL

We visualize our scene graph mapping results on several samples given ground-truth bounding boxes. The visualization results are shown in Fig. 4. We Further visualizes some failure cases in Fig. 5. In the first failure case, the model gets confused between "book" and "paper", while in the second case, the model fails to distinguish different plates. In Fig 6, we show that when there are multiple objects with the same category in the image, our model is able to distinguish them through object-wise relations.

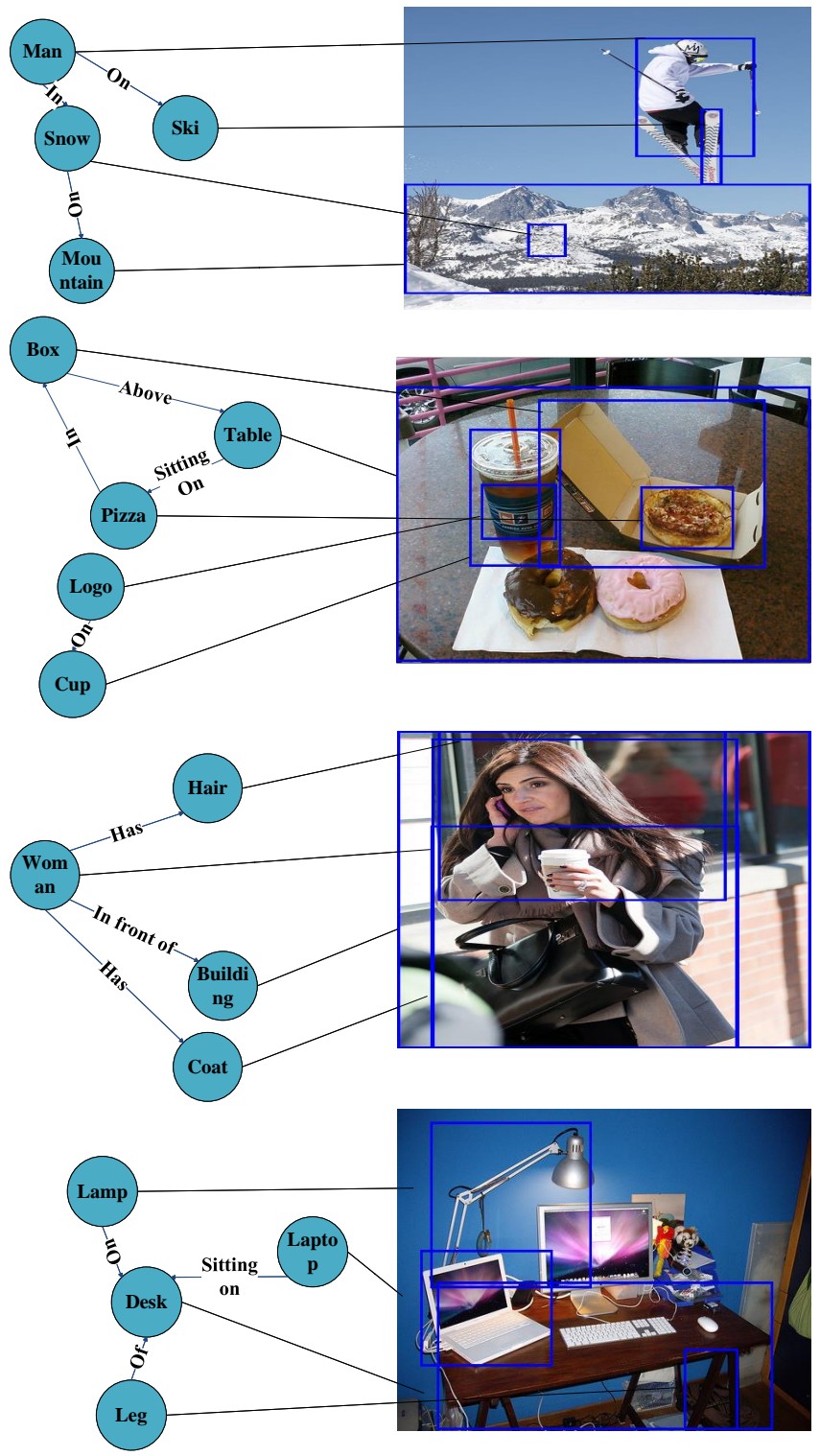

Figure 4: The visualization of the mapping found by our model given ground-truth bounding boxes.

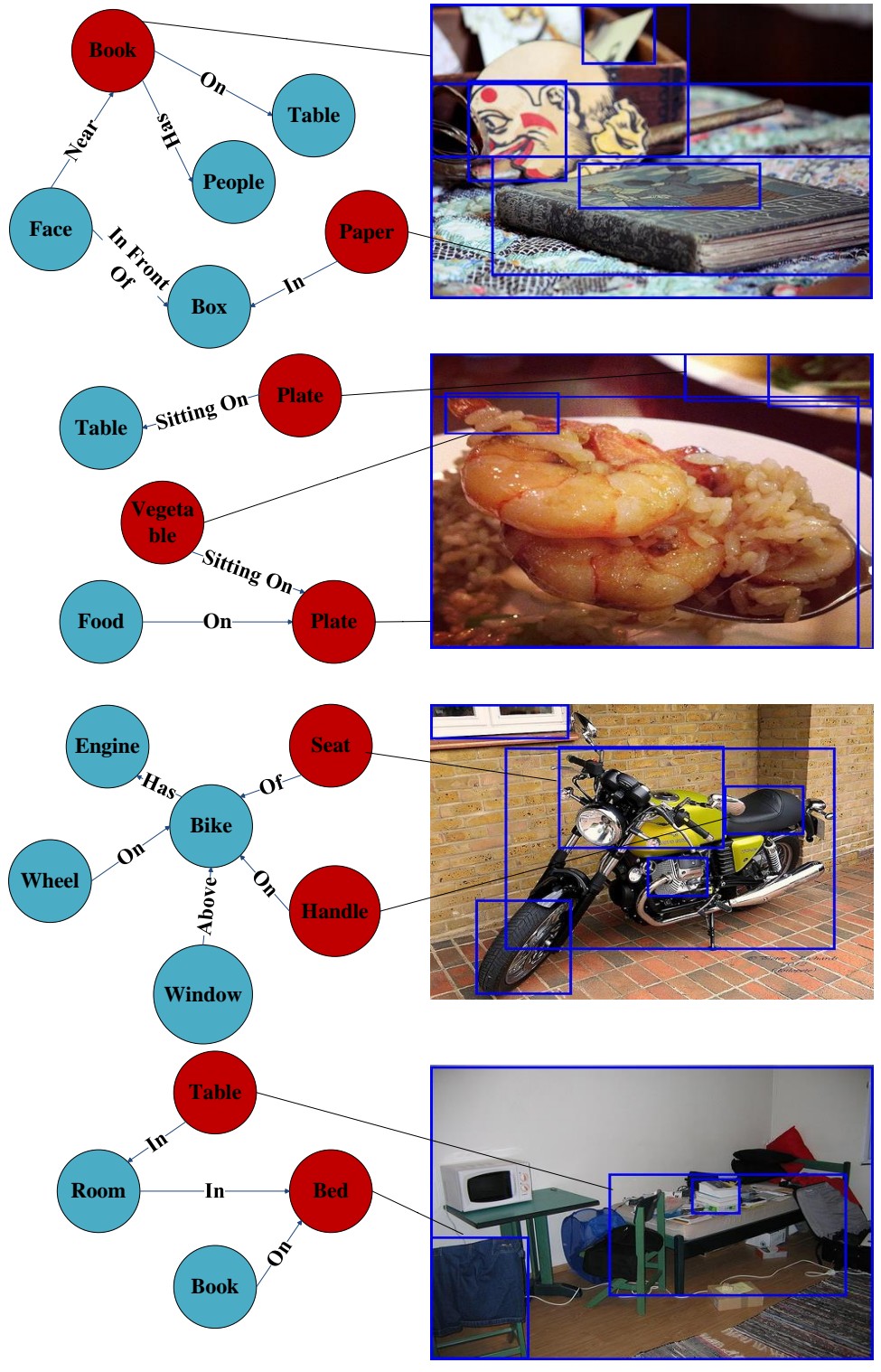

Figure 5: Visualization on Failure Cases of Our Model

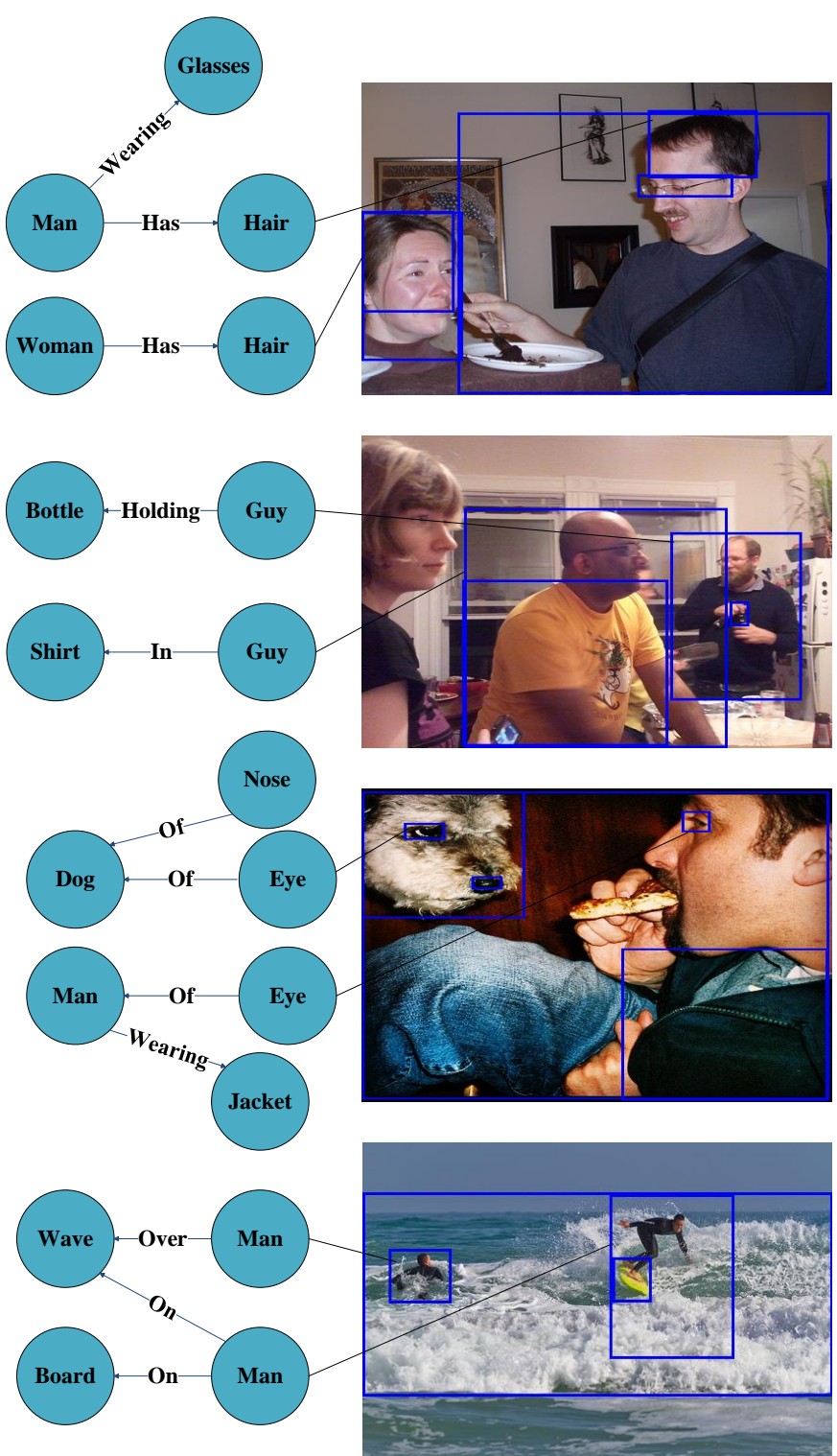

Figure 6: Visualizations on Ambiguous Objects with Relations

