# OpenReview forum: "Weakly Supervised Scene Graph Grounding"
_ICLR.cc/2021/Conference — Reject_

### Official Review · AnonReviewer3 · 2020-10-13
**An interesting and crucial task, but the novelty of idea is limited.**

**Rating:** 5
**Confidence:** 5

**Review:**

###############################
Summary:
This paper focuses on a new scene understanding task: Weakly-Supervised Scene Graph Grounding. Given the image and a paired scene graph, this task needs to link each scene graph node to an object in the image. Specifically, it regards this problem as a bipartite graph matching problem, and designs a cost function considering both object-class similarity and relation-based similarity. Experiments on two datasets (Visual Genome and Visual Relation Detection) have demonstrated the effectiveness.

################################
Advantages:
+ The paper is the first work to explore the task: Weakly-Supervised Scene Graph Grounding. This task can serve as a crucial preprocessing step for weakly-supervised scene graph generation: After solving this grounding task, the weakly-supervised scene graph generation task can be easily transferred to fully-supervised scene graph generation by regarding the grounding results as ground-truth annotations.

+ The whole paper is well-written and all notations are quite clear.

#################################
Weaknesses:

- The novelty of the idea for weakly-supervised scene graph grounding is limited. It regards this grounding task as a bipartite graph matching problem, which is very similar to the existing work for weakly-supervised scene graph parsing (ie, VSPNet).

- Although this paper is the first work to explore weakly-supervised scene graph grounding task, the task is very similar to the existing weakly-supervised phrase grounding task.

- The comparisons between the proposed method and baselines are not fair. For example, in Table 1, for the baseline WS-VRD [Baldassarre et al., 2020] model, it only utilizes predicates as input for weakly-supervised learning, while the proposed model uses the whole scene graph as input.

- The details of some experiments are not clear. For example, for the compared baseline VSPNet [Zareian et al., 2020a], the authors use two variants of models (VSPNet-v1 and VSPNet-v2), but the details of these two variants are not clear, ie, how to change an scene graph parsing model (VSP) for the scene graph grounding task.

---

> ### Author Response · Authors · 2020-11-23
> **Responses for AnonReviewer3**
>
> Thank you for your insightful review. Below are our response to your questions regarding the experiments setting:
>
> > The novelty of the idea for weakly-supervised scene graph grounding is limited. It regards this grounding task as a bipartite graph matching problem, which is very similar to the existing work for weakly-supervised scene graph parsing (ie, VSPNet).
>
> One key difference between our model and VSPNet is that while VSPNet computes matches over two generated graphs, we further process those graphs into embedding vectors. Furthermore, VSPNet does not consider weakly scene graph grounding separately, while its design limits the ability to learn an individual grounding system. Yet our experiments show that, with our proposed explicit scene graph grounding system, we can further achieve better grounding results as well as parsing results.
>
> > Although this paper is the first work to explore weakly-supervised scene graph grounding tasks, the task is very similar to the existing weakly-supervised phrase grounding task.
>
> Though our proposed task is close to phrase grounding, the inputs of those two tasks are different. In our setting, we expect the input scene graph to contain objects and pairwise relations, while phrase grounding needs a detailed description for example, “ a man in red”. In phrase grounding, the number of relationships associated with one noun is usually not large. For example, a phrase may only mention “a man in house”. But in scene graphs, one noun may be connected to many other nouns. It is hard and unnatural to adapt existing weakly supervised grounding models like GroundR on this task. Thus, we need a model that can aggregate multiple relationships.
>
> > The comparisons between the proposed method and baselines are not fair. For example, in Table 1, the baseline WS-VRD [Baldassarre et al., 2020] model, it only utilizes predicates as input for weakly-supervised learning, while the proposed model uses the whole scene graph as input.
>
> The design of the WS-VRD model does not support taking the whole graph as input. That’s one of the reasons why we consider designing a grounding model with the ability to read a whole graph as input since it carries more information. This is also a reason why we also include VSPNet as a baseline. VSPNet can take the whole graph as input in the graph alignment process. Our leading performance compared with both VSPNet and WS-VRD indicates that (1) enabling the model to read the whole graph as input is necessary and (2) our model works better in grounding with the whole graph input.
>
> > The details of some experiments are not clear. For example, for the compared baseline VSPNet [Zareian et al., 2020a], the authors use two variants of models (VSPNet-v1 and VSPNet-v2), but the details of these two variants are not clear, ie, how to change a scene graph parsing model (VSP) for the scene graph grounding task.
>
>
>
> VSPNet has a graph alignment module that solves graph matching in an iterative manner. In each iteration, the module can alternatively select starting from mapping edges or mapping nodes. We denote the mode starting from mapping edges as VSPNet-v1 and the one starting from mapping nodes as VSPNet-v2. This can be found in the Baseline paragraph in the Experiment section. As for the detailed process of how to apply VSPNet on scene graph grounding, we discussed the details in the fourth paragraph of the Introduction Sections. Although VSPNet is a model for VSP, it has a grounding module that can map the nodes to visual regions.

---

### Official Review · AnonReviewer1 · 2020-10-25
**see comments**

**Rating:** 4
**Confidence:** 5

**Review:**

Summary:
This paper presents a method on the task of Scene Graph Grounding, where the object and (object, object) relationship ground-truth labels are only at the image-level but not localized, i.e., instance-level. The core of the proposed method is the Wasserstein distance minimizer (aka, Earth Mover's Distance) for two object pair sets: one is from visual detection, and the other is from the textual nouns. The initial cost for each set is initialized by MIL probabilities. Experiments demonstrate a considerable performance boost over a SOTA (Zareian et al., 2020b)

Strength:
1. Strong performance in Table 1 (I do not mean that the experiments are comprehensive, see weakness below)


Weakness:
1. The motivation is not clearly justified in the Introduction. For example, for readers not familiar with this task, the authors may want to provide a clear illustrative example of this task, to show why scene graph grounding is challenging and significant. Moreover, compared to (Zareian et al., 2020b), whom the authors are targeting, in fact, I see no obvious weakness in their method and how you resolve it.

2. I am not objecting to use the simple EMD to solve weakly supervised matching problems, as it is straightforward. However, I feel that the authors are just using the right tool to do the right job, without convincing the readers why this tool is sufficient and necessary, especially compared to many other possible tools, such as using GNN, (Zareian et al., 2020b), MIL (Zhang et al. 2017b), and graphical models (Justin et al, Image Retrieval with Scene Graphs, CVPR'15).

3. The overall pipeline is not clearly illustrated. For example, after the EDM matching, how to obtain the final scene graph alignment?

4. Experiments are not comprehensive. First, I'd like to see visual qualitative examples of the success and failures of grounding, especially for improving weakly supervised SGG. Second, ablations such as using different metrics for the matrices are missing.

---

> ### Author Response · Authors · 2020-11-23
> **Response for AnonReviewer1**
>
> > The motivation is not clearly justified in the Introduction. For example, for readers not familiar with this task, the authors may want to provide a clear illustrative example of this task, to show why scene graph grounding is challenging and significant. Moreover, compared to (Zareian et al., 2020b), whom the authors are targeting, in fact, I see no obvious weakness in their method and how you resolve it.
>
> VSPNet aims at weakly supervised scene graph parsing and does not consider weakly supervised scene graph grounding as an individual task. Yet in our paper, we argue that weakly supervised scene graph grounding is worth being considered independently. We show that, by grounding scene graph nodes prior to parsing, the performance on scene graph parsing can be improved, regardless of whether proposals come from RPN or ground truth.
>
> > I am not objecting to use the simple EMD to solve weakly supervised matching problems, as it is straightforward. However, I feel that the authors are just using the right tool to do the right job, without convincing the readers why this tool is sufficient and necessary, especially compared to many other possible tools, such as using GNN, (Zareian et al., 2020b), MIL (Zhang et al. 2017b), and graphical models (Justin et al, Image Retrieval with Scene Graphs, CVPR'15).
>
> To our knowledge, GNN is more practical than Graphical Model because it has much more trainable parameters. Besides, MIL only focuses on relations between adjacent nodes, or in other words, local relations. So it misses the overview of the whole network. That is why we adopt GCN and embedding distance matching as our approach. Actually, in the experiment sections, we discussed the performance comparison between the MIL-based model and our model on weakly-supervised scene graph parsing in Section 6 (Table 4 and 5). We did not apply the implementation in Zhang et al. 2017b because their model does not support external proposals. As for graphical models, the authors did not release the code.
>
> > The overall pipeline is not clearly illustrated. For example, after the EDM matching, how to obtain the final scene graph alignment?
>
> After the EDM matching, each scene graph node is associated with one proposal of the highest similarity score. And thus the correspondence between graph nodes and object proposals has already been established.
>
> >Experiments are not comprehensive. First, I'd like to see visual qualitative examples of the success and failures of grounding, especially for improving weakly supervised SGG. Second, ablations such as using different metrics for the matrices are missing.
>
> We have visualized the success and failure grounding in the updated Appendix (Section E). The successful cases, together with Table 3 in the revised draft, verify that our model can learn to distinguish nodes of the same object class based on relational information. The failure cases also show some limitations of only considering object class and relations. Sometimes the model may simultaneously wrongly ground both the subject and object of a triplet. In this situation, it is hard for the model to correct the grounding with relational information. For example, in the second failure case, the model wrongly ground the “plate” and the “vegetable” on it. Thus, in future work, we may consider introducing attribute information. As for the discussion on the two different types of metrics (object-based and relational metric), we included them in Table 2 and Table 3 in the revised draft. As we can see, the object-based metric is not adequate to distinguish the nodes of the same class. By adding the relational metric, we can distinguish them better. A detailed discussion can be found in Sec 5.3. As for the different metrics, we show the results of different types of metrics (with or without relational information) in Table 2, and our discussion can be found in Sec 5.3. Our designing of the relational metric significantly improves the grounding accuracy, especially on the nodes with ambiguous nodes (nodes with the same object class) in the same graph.

---

### Official Review · AnonReviewer4 · 2020-10-26
**New problem, interesting findings, valuable insights.**

**Rating:** 7
**Confidence:** 5

**Review:**

This paper introduces a new problem called weakly supervised scene graph grounding, which is potentially very useful but barely studied in the literature. Given a dataset of images labeled with scene graphs, but without bounding box annotation, the task is to learn to align each node of a given scene graph to the corresponding object in the image. This is useful for weakly supervised training of scene graph generators, as well as other applications such as multimodal grounding. The authors clearly define the problem and propose an evaluation setting for it. They also propose a new method for this task and compare to existing baselines. Their findings are interesting, novel, and potentially influential for future work.

Nevertheless, there are some confusions/weaknesses that can be improved in the rebuttal:

1. There is insufficient insight about the proposed method and whether/why it works. Particularly, the authors first train a weakly supervised object recognition model using a multiple instance learning framework (Eq. 3), then use that to align ground truth nodes to objects (Eq. 9), and then use this alignment as a ground truth for training the graph neural networks (Eq. 10). However, the ground truth alignment produced by the object recognition model only takes into account object classes and not their relations, while the goal of the graph neural networks is to incorporate relations to disambiguate between instance of the same class. Hence, the ground truth is inadequate to learn the task and the model may just learn to ignore relations and use objects alone to do the alignment. Experiment results (table 2) confirm this concern to some extent, as the ablation without relational metric (no graph neural net) achieves very close performance to the proposed model. Therefore, it is essential for the authors to justify their proposed model e.g. using extra experiments/analyses/visualizations, to prove that the graph neural networks indeed does learn to take relations into account for grounding objects.

2. The writing of the paper can be improved. It is not very clear, except from Algorithm 2, that the training framework has 2 phases: training the object recognition model, and then training the graph comparison model based on that. The authors may add an overview to explain this. There are also notation issues and confusions in the equations. For instance, the definition of U, N, and Y below Eq 3 is confusing. Also it is not clear what is the input set to \sigma (pooling). The definition of \pi in Eq. 8 is also not precise enough. And in Eq. 10, the second equation is confusing, maybe the \lambda inside the sum should be removed. Furthermore, Section 6 should be elaborated more. For instance, it should be made clear that this makes the training a 3-step process: learning object recognition, then learning the graph metric, and then using that to learn the scene graph generation model.

3. Experiments provide new findings and useful insights, but they can be improved. For instance, it is not clear why, even though the proposed alignment method is far better than the VSPNet baseline in table 1, its effect on the scene graph generation performance is small (table 3). Moreover, the SGCls and PredCls evaluation metrics are both unrealistic metrics that assume ground truth bounding boxes are available during inference. The authors are encouraged to report SGDet, to show how effective the proposed method is in real-world settings.

Despite the suggested improvements in experimentation, writing, and justification of the method, this paper carries a significant value for the community, as it calls attentions to a new task, that is barely studied, but has potential applications. The empirical findings of the paper provide new insights that do not exist in the literature, and can directly guide future work. Hence, my recommendation is to accept this paper.

The authors are strongly encouraged to address the concerns mentioned above. I also recommend resolving grammar mistakes throughout the paper, although that should not affect the final decision. Here are some more suggestions to improve the paper:

1. The authors are encouraged to more extensively justify why this task is important and difficult. They did provide an example of a person riding a horse while another person is riding a bike, but that single example may not be convincing enough for others to pursue further research in this direction. The authors may provide some statistical analysis of how many difficult grounding cases exist in the dataset, what makes these cases difficult, and how improving the grounding can improve downstream tasks such as scene graph generation.

2. There are some recent papers on visual grounding by incorporating relationships between objects, such as [1]. I recommend including a comprehensive review of such papers and discussing their connections and differences with this work.

[1] He, Chuanzi, Haidong Zhu, Jiyang Gao, Kan Chen, and Ram Nevatia. "CPARR: Category-based Proposal Analysis for Referring Relationships." In Proceedings of the IEEE/CVF Conference on Computer Vision and Pattern Recognition Workshops, pp. 948-949. 2020.


######## Post-Rebuttal Updates:

The authors have addressed most of my concerns, and I appreciate their effort. I am not convinced that early-stopping alone can help the model learn to utilize relation information, when the target ground truth does not require the model to do so. However, I recommend accepting this paper as it introduces a new task and presents strong results.

---

> ### Author Response · Authors · 2020-11-23
> **Response for AnonReviewer4**
>
> Thank you for the questions and insightful suggestions! For the suggestions about writing, we have addressed them and updated the draft. Below are our responses to other questions and concerns about the proposed method and experiment results:
> > “However, the ground truth alignment produced by the object recognition model only takes into account object classes and not their relations, while the goal of the graph neural networks is to incorporate relations to disambiguate between instances of the same class. Hence, the ground truth is inadequate to learn the task and the model may just learn to ignore relations and use objects alone to do the alignment.”
>
>  Thank you for your questions. Intuitively, if the GNN has gotten overfitted on the output from object-class based metric, it will ignore all relations and only apply the object. To avoid overfitting, we apply early-stopping via a small validation set. We conduct an experiment that shows our model can use relational information to distinguish the nodes with the same object class. The new results are included in Table 3 in the revised version of the paper. The reason why “the ablation without relational metric (no graph neural net) achieves very close performance to the proposed model” is that relational information is crucial for a fraction of the nodes rather than all of them. For the nodes without ambiguous nodes (nodes of the same object class), object class information is adequate for the grounding. From Table 3, we can see that the relational metric significantly increases the grounding accuracy of the ambiguous nodes with relations (by 5%). Also, we visualized some cases in Figure 6 of the updated draft, showing that the ambiguous nodes can be distinguished by our model.
>
>
>
>
>
> >Experiments provide new findings and useful insights, but they can be improved. For instance, it is not clear why, even though the proposed alignment method is far better than the VSPNet baseline in Table 1, its effect on the scene graph generation performance is small (table 3). Moreover, the SGCls and PredCls evaluation metrics are both unrealistic metrics that assume ground truth bounding boxes are available during inference. The authors are encouraged to report SGDet, to show how effective the proposed method is in real-world settings.
>
> The improvement of our grounding methods on scene graph generation for VSPNet is small because the scene graph parsing is a challenging task itself even with ground truth labels. For instance, the Recall@50 of VSPNet FS (in Table 3 of the original paper, Table 4 in the revised version) only lead the VSPNet WS with 1 percent, while the VSPNet FS is trained with the ground truth grounding (in other words, with 100% accuracy). Meanwhile, our grounding model, although significantly outperforming the baselines, is with 50.8% accuracy (less than 10 percent leading). We can also note that on other metrics, like PredCLS, our grounding brings more significant improvement. As for the SGDet metric, we really appreciate your suggestion and we have uploaded the results in the Table 5 rebuttal version of the draft. Our grounding model also improves the performance of SGDet (we use the name of "SGGen" to keep consistent with VSPNet).
>
> >The authors are encouraged to more extensively justify why this task is important and difficult. They did provide an example of a person riding a horse while another person is riding a bike, but that single example may not be convincing enough for others to pursue further research in this direction. The authors may provide some statistical analysis of how many difficult grounding cases exist in the dataset, what makes these cases difficult, and how improving the grounding can improve downstream tasks such as scene graph generation.
>
> Thank you for your suggestions! We have included dataset analysis of the ambiguous nodes difficult for (grounding) in Table 3 of the updated draft.

---

### Official Review · AnonReviewer2 · 2020-10-28
**Problem is new, but the experiment is weak**

**Rating:** 5
**Confidence:** 4

**Review:**

**Summary**:
This paper proposes a weakly supervised scene graph grounding method. The setting of this paper is new and interesting, and I think this task would be beneficial to downstream tasks. The main contribution of this method is the mapping between visual objects and nodes. To bridge the gap between them, the authors propose two metrics and learn their model on a weakly-supervised dataset. The ablation study shows the effectiveness of each contribution, and they further apply the method to weakly supervised scene graph parsing.

**Pros**:
- This paper proposes the weakly supervised scene graph grounding task and formulates the mapping between scene graph and objects as a minimum match problem on a bipartite graph instead of graph alignment.
- They train their model with two metrics (object-class based and relation-based) and search the minimum match in an iterative manner.
- The visualization shows that nodes and objects can be well aligned.

**Cons**:
- What's the design of pool-based multiple instance learning? Can you provide more details?
- The updating algorithm of the scene graph is not new; it is close to the graph encoding process; it updates the representations according to its neighbor.
- In fig. 3, the authors mentioned that the boxes are ground-truth, while they also mentioned that the visual objects are extracted by off-the-shelf Faster-RCNN.
- For VG and VRD, in my opinion, the objects have been semantically aligned with scene graph nodes since the objects are detected by the pre-trained detector (OpenImage). Why not use a scene graph generated from captions as the input.
- Why the $\lambda$ for VG and VRD is different, how to get those values?
- As for visualization, the authors only show the scene graphs with the same objects. They should also show the images with objects, not in the scene graph.
- The writing needs to be improved; for example, Eq. 11 should be exp(-d(u,v)).

---

> ### Author Response · Authors · 2020-11-23
> **Response for AnonReviewer2**
>
> Thank you for your suggestions on improving the experiment section and the writing of this paper. We have updated the draft and fixed the typos. Belows are our responses to your questions on the experiments setting:
> Cons:
> > What's the design of pool-based multiple instance learning? Can you provide more details?
>
> We provided a more detailed description in the appendix of the revised draft. It is in the A section of the Appendix.
>
> > In fig. 3, the authors mentioned that the boxes are ground-truth, while they also mentioned that the visual objects are extracted by off-the-shelf Faster-RCNN.
>
> In our framework, the object features are extracted via an off-the-shelf Faster-RCNN, while the object bounding boxes can be either ground truth or proposals generated by off-the-shelf Faster-RCNN. We reported the numerical grounding results of both settings in the experiment section and provided the visualization of the ground truth box setting.
>
> > For VG and VRD, in my opinion, the objects have been semantically aligned with scene graph nodes since the objects are detected by the pre-trained detector (OpenImage). Why not use a scene graph generated from captions as the input.
>
> The object detector is pre-trained on OpenImage dataset, whose label space is different from the VG and VRD dataset. This is the reason why we need to learn the object class based metric to align the object category semantic
>
> > Why the λ for VG and VRD is different, how to get those values
>
> We conduct hyper-parameter search on the validation split. From Table 1 and 2 we can see that VRD is a relatively “simpler” dataset (most of the baselines and variant models achieve higher scores on VRD dataset) and the object-class based metric has already achieved a high performance (66.7). Thus, the relationship based metric is not so important and can be assigned with a smaller weight.

---

### Decision · Program_Chairs · 2021-01-07
**Final Decision**

**Decision:**

Reject

**Comment:**

This paper presents work on scene graph grounding under weak supervision.  The reviewers appreciated the consideration of this task and formulation of a solution for it.  However, concerns were raised over the importance of this weakly-supervised grounding task, how it addresses challenges in previous methods, the empirical evaluation, insights obtained, motivation, and clarity of exposition.  After reading the authors' response, the subsequent discussion and reconsideration resulted in a sense that while the task is new, the overall contribution and remaining questions over empirical evaluation mean the paper is not yet ready for publication at ICLR.